# Surgical and Infectious Complications Following Kidney Transplantation: A Contemporary Review

**DOI:** 10.3390/jcm14103307

**Published:** 2025-05-09

**Authors:** Kazuaki Yamanaka, Yoichi Kakuta, Shigeaki Nakazawa, Kenichi Kobayashi, Norio Nonomura, Susumu Kageyama

**Affiliations:** 1Department of Urology, Shiga University of Medical Science, Otsu 520-2192, Japan; den4low@belle.shiga-med.ac.jp (K.K.); kageyama@belle.shiga-med.ac.jp (S.K.); 2Department of Urology, Osaka University Graduate School of Medicine, Osaka 565-0871, Japan; kakuta@uro.med.osaka-u.ac.jp (Y.K.); nakazawa@uro.med.osaka-u.ac.jp (S.N.); nono@uro.med.osaka-u.ac.jp (N.N.)

**Keywords:** renal transplantation, clinical complication, transplant surgery

## Abstract

Kidney transplantation significantly improves outcomes in patients with end-stage renal disease; however, postoperative complications remain a substantial concern. This review summarizes the incidence, risk factors, and management strategies for common complications after kidney transplantation. Reported incidence varies widely due to differences in definitions, diagnostic methods, and study designs. Ureteral stenosis occurs in 2.8–18.0% of recipients, vesicoureteral reflux in 0.5–86%, and urinary leakage in 1.1–7.2%. Lymphatic complications, including lymphocele and lymphorrhea, range from 0.6% to 35.2%, with one-third of complications requiring intervention. The incidence of urinary tract infections ranges from 20 to 43%, while asymptomatic bacteriuria is reported in up to 53% of recipients. Surgical site infections have a median incidence of 3.7%, and incisional hernias develop in 2.5–10% of cases, depending on follow-up duration. Vascular complications affect approximately 10% of recipients, with renal artery stenosis and thrombosis being the most prevalent. Neurologic complications, such as femoral nerve palsy and immunosuppression-related neurotoxicity, though less frequent, can impair recovery. Management strategies vary depending on severity, ranging from observation to surgical intervention. Preventive measures—including optimized ureteral stenting protocols, early catheter removal, careful immunosuppression, and appropriate antimicrobial use—play a crucial role in reducing complication risk. Despite advances in transplantation techniques and perioperative care, these complications continue to affect graft survival and patient outcomes. Further research is needed to standardize definitions and establish evidence-based protocols.

## 1. Introduction

Kidney transplantation has been demonstrated to be an effective renal replacement therapy for patients with end-stage renal disease, significantly improving both survival rates and quality of life [1]. However, the occurrence of perioperative complications not only reduces graft survival but also worsens the prognosis of kidney transplant recipients [2,3,4]. To standardize the assessment of surgical complications, the Clavien–Dindo classification is widely used. This system stratifies complications according to the invasiveness of the treatment required and has been increasingly applied in the field of kidney transplantation [5]. Dagnæs-Hansen et al. employed this classification to systematically analyze both short- and long-term postoperative complications in a large institutional cohort, reporting that 60% of kidney transplant recipients experienced at least one complication within 30 days after transplantation. Furthermore, they identified previous transplantation, advanced age, underweight status (BMI < 18.5), and intraoperative blood loss as significant risk factors for severe complications graded as Clavien–Dindo > 2. In particular, each 100 mL increase in intraoperative blood loss was associated with a significantly higher risk of CD > 2 complications (odds ratio: 1.11; 95% CI: 1.01–1.21; *p* = 0.032) [6]. These findings underscore the importance of thorough preoperative risk assessment, meticulous surgical technique, and the early detection and appropriate management of complications using objective criteria.

This review provides an overview of major complications that frequently occur and have a significant impact on the clinical course of kidney transplant recipients. Specifically, it discusses urological complications (ureteral stenosis, vesicoureteral reflux (VUR), and urinary leakage), lymphatic disorders (lymphocele and lymphorrhea), infections (urinary tract infections (UTIs) and surgical site infections (SSIs)), vascular complications, neurologic complications, and incisional hernia. Given the continuous advancements in transplantation medicine, including improvements in diagnostic techniques and treatment approaches for complications, this review particularly focuses on recent studies published since 2000.

## 2. Urinary Complications

The occurrence of ureteral complications has been reported to significantly reduce graft survival. In particular, recipients who develop ureteral stricture exhibit markedly shorter graft and overall survival [2]. Among anatomical considerations, special attention should be paid to the peri-ureteral tissue surrounding the distal ureter and renal hilum—specifically the so-called “golden triangle”. This critical area is traditionally defined as the triangular region bordered by the renal hilum, the lower pole of the kidney, and either the right renal vein–inferior vena cava (IVC) junction or the gonadal–left renal vein junction [7]. It contains vital arterial branches, such as the lower polar artery, which plays a key role in maintaining adequate blood supply to the distal ureter. Surgical trauma or excessive dissection in this zone during graft retrieval or backbench preparation may compromise ureteral perfusion, leading to postoperative complications such as urinary leakage or ureteral stricture. Moreover, recent laparoscopic and minimally invasive donor procedures have emphasized the importance of an even wider “safety triangle,” defined as the area between the lower pole of the kidney and the gonadal vein, extending to its confluence with the renal vein. Preserving the peri-hilar and peri-ureteral vascularized fat within this expanded zone is essential to maintain ureteral vascular integrity and minimize the risk of postoperative urological complications. These anatomical insights underscore the importance of meticulous surgical technique and careful handling of the peri-ureteral region during graft procurement and preparation in order to preserve ureteral viability and optimize graft outcomes.

Regarding anastomotic techniques, Alberts et al. conducted a review comparing commonly performed ureterovesical anastomotic techniques, including the Lich–Gregoir extravesical technique, the Politano–Leadbetter intravesical technique, and the U-stitch technique. Their findings indicated that the Lich–Gregoir extravesical technique was associated with the lowest incidence of urological complications [8]. Furthermore, a separate randomized controlled trial (RCT) reported that the extravesical technique was associated with reduced operative time and a lower incidence of UTIs. Additionally, a modified approach to the conventional Lich–Gregoir extravesical technique has been described, incorporating several refinements, including mobilization of the bladder, extended spatulation of the ureter, inclusion of the bladder mucosa along with the detrusor muscle layer in the ureteral anastomosis, and the use of a right-angle clamp at the ureteral orifice to prevent stenosis. This modified technique has been reported to reduce the incidence of urological complications to as low as 1.4% [9]. A comparison between primary ureteroureterostomy and conventional ureteroneocystostomy has demonstrated that the incidence of VUR and UTIs is lower in the ureteroureterostomy group, whereas the occurrence of ureteral obstruction is higher. However, no significant difference has been observed between the two techniques in terms of the overall incidence of urological complications [10,11]. Although no significant difference in the overall incidence of urological complications has been observed between ureteroneocystostomy and ureteroureterostomy, ureteroneocystostomy is generally preferred as the initial approach. This preference is due to the possibility of salvaging complications via a secondary ureteroureterostomy if needed. However, low-capacity bladder, which often occurs in recipients undergoing kidney transplantation after long-term dialysis, has been associated with an increased risk of urological complications. In such cases, primary ureteroureterostomy may be considered a reasonable alternative [12]. Non-absorbable sutures used for ureteroneocystostomy have been associated with urinary stone formation; therefore, absorbable sutures are recommended to minimize this risk [13].

Regarding the utility of stents, the European Association of Urology guidelines strongly recommend their placement, stating the following: Use transplant ureteric stents prophylactically to prevent major urinary complications [14]. A Cochrane review concluded that universal prophylactic ureteric stenting, compared to no stenting, likely reduces the incidence of major urological complications, irrespective of the duration of stent placement. This conclusion was based on data from 11 studies involving 1834 participants (RR: 0.30, 95% CI: 0.16–0.55; *p* < 0.0001; I^2^ = 16%; moderate-certainty evidence; number needed to treat = 17) [15]. Regarding the risk of UTIs associated with ureteric stent placement, no significant difference has been observed with short-term stenting. However, prolonged stent placement may increase the risk of UTI development [16,17,18]. However, it has been reported that the use of trimethoprim-sulfamethoxazole (TMP-SMX) prophylaxis eliminates the significant difference in UTI incidence between patients with and without ureteric stents [19]. A systematic review and meta-analysis recommended a stent removal period of three weeks, as this duration did not increase the risk of UTIs and showed no significant difference in the incidence of ureteral stricture or urinary leakage [20].

### 2.1. Ureteral Stenosis

#### 2.1.1. Etiology

Ureteral stenosis after kidney transplantation is a significant postoperative complication that can threaten graft function and survival. Numerous studies have suggested that ureteral stenosis increases the risk of graft dysfunction and loss [2,21]. The incidence of ureteral stenosis varies depending on study design, patient population, and the definition of stenosis. However, it has been reported to occur in a median of 3.8% (range: 2.8–18.0%) of kidney transplant recipients [2,11,22,23,24,25,26,27,28,29,30]. The majority of cases occur within three months to one year post-transplant [2,22,26].

The diagnosis of post-transplant ureteral stenosis is typically made using imaging modalities such as ultrasound and computed tomography scans, with the detection of hydronephrosis often serving as the initial indicator. While stenosis can occur at any site along the ureter, it is most commonly reported at the distal ureter and the ureterovesical anastomosis [2,22,31,32]. The anastomotic site, where the ureter is connected to the bladder, is particularly prone to stenosis due to surgical manipulation and compromised blood supply.

Causes of extrinsic ureteral compression include lymphocele and hematoma, with lymphocele being the most frequently reported cause [30,33].

#### 2.1.2. Risk Factors

The risk factors for ureteral stricture can be broadly classified into three categories: donor-related factors, recipient-related factors, and surgical factors.

Recipient-related factors: Several studies have reported a higher incidence of ureteral complications in male recipients; however, the underlying reason remains unclear [2,27]. Delayed graft function (DGF), in which the transplanted kidney does not function immediately, increases the risk of ureteral stenosis [2,34]. This is thought to be due to the fact that DGF is partly caused by inadequate blood flow to the graft, which may also reduce blood supply to the ureter [35]. Some reports suggest that rejection and UTIs can cause inflammation and scarring of the ureter, potentially increasing the risk of stenosis [14]. Notably, BK virus infection, which is more common in immunocompromised patients, has also been associated with ureteral stenosis [36]. However, large-scale studies have not clearly demonstrated rejection or UTIs as definitive risk factors for ureteral stenosis.

Donor-related factors: As donor age increases, the likelihood of arteriosclerosis and other vascular changes affecting the ureter also rises. This may lead to reduced blood flow to the ureter post-transplant, thereby increasing the risk of stenosis [2,34]. Some reports suggest that donor creatinine levels may also be a risk factor, potentially reflecting impaired donor blood flow [26]. Additionally, arterial reconstruction of the donor kidney has been reported as a risk factor, with relative ischemia of the ureter due to insufficient arterial blood supply being suggested as the underlying cause of ureteral leakage and stenosis [27].

Surgical-related factors: Intraoperative surgical manipulation can damage the feeding vessels of the ureter, leading to ischemia and increasing the risk of stenosis. To assess blood flow, the use of intravenous indocyanine green fluorescence imaging during surgery has been reported to be beneficial [37]. Although it was previously thought that a shorter ureter length could increase tension, impair blood flow, and raise the risk of stenosis, recent reports indicate that ureter length is not associated with a higher incidence of urological complications [25]. Additionally, a study examining peri-ureteric connective tissue preservation in living-donor kidney transplantation found that preservation of peri-ureteric tissue within living-donor kidney transplantation was not independently associated with urological complications [29]. Apel et al. reported that predictive nomograms calculated using donor creatinine, residual diuresis, and body mass index were developed to estimate an individual patient’s risk of developing ureteral stenosis and the probability of remaining stenosis-free for 12 months after transplantation [26].

#### 2.1.3. Treatments

Various treatment options are available for ureteral stricture, including stent placement, balloon dilation, endoureterotomy, and surgical repair. The choice of treatment should be individualized based on the severity, location, underlying cause of the stricture, and the patient’s overall condition. Additionally, some cases have been reported to resolve spontaneously; however, clear criteria for selecting conservative management are lacking [25].

Stent placement: A stent is placed at the site of the stricture to maintain urinary drainage. This method is sometimes used in combination with balloon dilation, but the stent itself can contribute to stricture recurrence, infections, or stone formation. If transurethral insertion is not feasible, percutaneous nephrostomy catheter insertion into renal grafts is an alternative approach. This technique has been confirmed to be safe, simple, and effective with a low complication rate [24,38].

Balloon dilation: This minimally invasive procedure involves inserting a balloon catheter into the stricture site and inflating it to expand the narrowed area. However, the restenosis rate is high, approximately 50%, often requiring multiple treatments or subsequent surgical repair [32,39].

Endoureterotomy: The Holmium:yttrium-aluminum-garnet (Ho:YAG) laser endoureterotomy has demonstrated high success rates and is considered an effective treatment for ureteral strictures of 10 mm or shorter and for cases where antegrade percutaneous balloon dilation has failed in kidney transplant recipients [40,41].

Surgical repair: If the aforementioned methods are ineffective or if the stricture is complex, surgical reconstruction is performed, which involves excising the stricture and reconstructing the ureter [31,42]. Native ureteropyelostomy and redo-ureterocystostomy can be performed via open surgery or laparoscopic surgery. A study reported that graft survival was significantly shorter in kidney transplant recipients who initially underwent minimally invasive treatment for ureteral stricture, compared to those without strictures. However, recipients who underwent initial open surgery for ureteral stricture had no significant difference in graft survival rates compared to those without strictures, suggesting that initial open surgery might be more beneficial for long-term graft prognosis [2]. Regarding the re-anastomosis of the transplant ureter to the bladder or native ureter, a three-year follow-up study found no significant difference in serum creatinine levels between the two approaches. However, native ureteropyelostomy significantly reduced the risk of pyelonephritis [43].

Since the incidence of ureteral stricture is relatively low, there is a lack of high-quality studies comparing surgical techniques, underscoring the need for further research in this area.

### 2.2. Vesicoureteral Reflux (VUR)

#### 2.2.1. Etiology

VUR after kidney transplantation is a condition in which urine flows backward from the bladder into the ureter and, in some cases, reaches the transplanted kidney. VUR is a relatively common complication post-transplantation, with reported incidence rates ranging from 0.5% to 86% [44]. This wide variation in incidence is attributed to differences in the definition of VUR and the screening criteria used across studies. Recent reports indicate that among all kidney transplant recipients, the median prevalence of VUR in those who develop symptomatic UTIs is 5.2% (range: 1.1–10%) [30,43,45,46,47,48,49,50,51,52]. However, in centers where voiding cystourethrography is routinely performed on all post-transplant patients, the reported prevalence of VUR exceeds 40% in many studies [53,54,55]. While some studies have found no significant difference in UTI incidence between patients with and without VUR [54], screening for VUR is recommended in cases of recurrent UTIs [44]. If left untreated, VUR may lead to recurrent UTIs and progressive kidney dysfunction. Notably, high-grade VUR is associated with an increased risk of renal impairment [47,51,55,56]. A study evaluating renal damage using DMSA scans reported that 69% of patients with both VUR and UTI had renal scarring, whereas scarring was rarely observed in patients without either condition [57].

#### 2.2.2. Risk Factors

Recipient-related factors: Long-term dialysis patients are at an increased risk of VUR due to a low-capacity bladder [58]. Although small-scale studies suggest an association between a low-capacity bladder and VUR, bladder compliance has not been shown to be a contributing factor [59]. It is believed that in a low-capacity bladder, a sufficient submucosal tunnel cannot be created during ureterovesical anastomosis, leading to an increased risk of VUR.

Surgery-related factors: The Lich–Grégoir (extravesical) and Politano–Leadbetter (intravesical) techniques are commonly used for ureterovesical anastomosis. A systematic review comparing the incidence of VUR between these two methods found no significant difference [8]. Additionally, the surgeon’s experience has been identified as a contributing factor in VUR development. Studies have reported higher rates of VUR when the procedure is performed by less experienced surgeons [54].

#### 2.2.3. Treatments

Post-transplant vesicoureteral reflux (VUR) can lead to recurrent urinary tract infections (UTIs) and an increased risk of graft dysfunction, making treatment necessary in many cases [44,45,60]. The selection of treatment for VUR depends on the severity of reflux, the presence of UTIs, and the overall condition of the patient [49].

Observation and Prophylactic Antibiotic Therapy: Observation is considered for asymptomatic and low-grade (Grade I–III) VUR, particularly in patients with good graft function or high surgical risk [44,46,55]. Studies suggest that low-grade VUR does not negatively impact long-term graft function or survival. However, in cases of recurrent UTIs, prophylactic antibiotic therapy may be used [49]. Recent reports on primary VUR indicate that while prophylactic antibiotics can reduce UTI risk, their long-term use may contribute to antibiotic resistance. Therefore, alternative approaches, such as probiotics, have been suggested as potential treatment options that require careful consideration [61,62,63,64].

Transurethral Endoscopic Surgery: For symptomatic mild-to-moderate VUR, endoscopic injection of bulking agents is a minimally invasive treatment option. While endoscopic treatment has a lower success rate and higher recurrence risk compared to open surgery, it is repeatable, associated with a low risk of complications, and less invasive [49,50,60,65]. Bulking agents such as dextranomer/hyaluronic acid copolymer or polydimethylsiloxane are injected submucosally at the ureterovesical junction, creating a bulge that reduces reflux.

Open Surgical Ureteral Reimplantation: Surgical reimplantation is considered in cases of high-grade VUR or failure of endoscopic treatment [44]. Techniques include ureteroureterostomy, in which the transplanted ureter is anastomosed to the native ureter, and pyeloureterostomy, where the renal pelvis of the transplanted kidney is anastomosed to the native ureter [45,66]. Open surgical treatment generally has a higher success rate than endoscopic treatment but also carries a higher risk of complications.

### 2.3. Urinary Leakage

#### 2.3.1. Etiology

Urine leakage is one of the most common early surgical complications following kidney transplantation. Reports since 2005 have indicated a median incidence of 2.7% (range: 1.1–7.2%) [2,11,24,25,27,28,29,30,67]. Although there are no large-scale studies specifically focused on post-transplant urine leakage, the most frequently affected site is the anastomosis, followed by leakage from the renal pelvis or ureter due to surgical techniques [24,68,69,70]. In most cases, urine leakage occurs within three months post-transplantation, with ureteral necrosis and anastomotic dehiscence being the primary causes [29,31]. Ureteral necrosis is most commonly caused by ischemia due to damage to the ureter’s blood supply during surgery [10,69,71]. To reduce the risk of ureteral necrosis, it is crucial to preserve the peri-ureteric connective tissue to maintain adequate blood supply to the distal ureter [31]. Diagnosis of urine leakage is typically performed by measuring creatinine levels in the drainage fluid [30]. Risk factors for urine leakage include advanced donor age, recipient age, diabetes, and the number of renal arteries [9,10,11,27,67,71,72,73].

#### 2.3.2. Treatments

The management of urinary leakage varies depending on its location, cause, and severity [31]. Treatment strategies are not based on solid scientific evidence but rather on clinical experience, leading to variations among institutions. The currently available evidence is primarily based on retrospective studies.

Conservative Treatment: Conservative treatment can be selected for cases of early and small-volume urinary leakage [14]. This approach aims to bypass urine away from the leakage site and achieve complete urinary decompression, thereby promoting healing [31]. The following methods are used: urethral catheter placement, percutaneous nephrostomy, JJ stent placement, or Drainage procedures. The catheters used in conservative treatment should remain in place until urinary leakage resolves.

Surgical Treatment: If conservative treatment fails to resolve urinary leakage, particularly in cases of massive extravasation or ureteral necrosis, surgical intervention is required [14,31]. The necrotic ureter is excised until viable tissue is reached, followed by reimplantation. If the remaining ureter is too short, the following techniques may be considered to achieve tension-free ureteral anastomosis: Ipsilateral ureteroureterostomy, Psoas hitch procedure, Boari flap procedure, or Ileal ureter substitution.

## 3. Lymphatic Disorders (Lymphocele and Lymphorrhea)

Lymphocele and lymphorrhea are relatively common complications after kidney transplantation. Reported incidence rates vary widely among studies, ranging from 0.6% to 35.2%, with approximately one-third of cases requiring invasive treatment [74,75,76,77,78]. Even in recent reports, there is considerable variability in incidence, likely due to differences in patient backgrounds, definitions of lymphocele and lymphorrhea, diagnostic methods, surgical techniques, postoperative management, and whether symptomatic or asymptomatic cases were included in the analysis.

The sources of lymphorrhea are considered to be either the recipient’s iliac lymphatic vessels, the donor kidney’s hilar lymphatics, or both [79]. Although usually asymptomatic, lymphorrhea can cause pain, swelling, infection, and kidney dysfunction due to compression of the graft or ureter. When distinguishing lymphatic fluid accumulation from urine leakage, measuring creatinine concentration in the accumulated fluid is a useful diagnostic approach. The median time to diagnosis of lymphocele formation is 29 days (IQR: 19–51) post-transplantation. Symptomatic lymphoceles requiring interventional treatment were managed at a median of 22 days (IQR: 8–55) after diagnosis. However, the development or treatment of lymphoceles does not appear to negatively impact long-term graft survival [78,80,81].

### 3.1. Risk Factors

Multiple factors contribute to the development of lymphatic complications, which can be categorized into recipient-related, surgical, and postoperative management factors (Table 1).

Recipient-related factors include diabetes [82], obesity [83], advanced age [81,84,85], a history of previous abdominal surgery [80,84], minimally invasive surgery, and rejection episodes [75]. Additionally, symptomatic lymphocele has been reported to be associated with DGF [78].

Surgical factors include aspects related to both the donor kidney and the recipient’s surgical procedure. One reported risk factor is the presence of multiple renal arteries in the donor kidney [86]. Regarding the recipient’s surgery, inadequate ligation of lymphatic vessels, lymphatic vessel injury from electrocautery, and the use of the external iliac artery for anastomosis have all been implicated in increasing the risk of lymphocele formation. These factors are believed to contribute to more extensive disruption of lymphatic drainage, making lymphocele formation more likely [87,88].

Postoperative management factors involve immunosuppressive therapy and drain placement. The use of tacrolimus (Tac), mycophenolate mofetil (MMF), Mammalian Target of Rapamycin inhibitors (mTORI), and high-dose steroid therapy has been reported as potential risk factors for lymphocele formation [33,74,81,82,89,90]. Regarding drain placement, conflicting reports exist. Some studies suggest that drain placement may help prevent lymphocele formation, while others indicate that it may actually increase the risk of lymphatic leakage [76,82,87]. One study found that while drain placement reduced the overall incidence of lymphocele, it did not affect the rate of symptomatic complications [91]. Although controversial, it is generally recommended to keep the drain in place until lymphatic output decreases sufficiently, ideally below 50 mL per day. However, prolonged drain placement increases the risk of infection [75,86].

**Table 1 jcm-14-03307-t001:** Risk factors for lymphatic complications.

	Risk Factors	Description	Reference
Patient-related factors			
	Diabetes mellitus	Delayed wound healing due to diabetes increases the risk of lymphocele formation.	[82]
	Obesity	Recipients with a high body mass index (BMI), especially those exceeding 30 kg/m^2^, have a higher risk of developing lymphocele.	[83]
	Advanced Age	Reduced healing capacity in both donors and recipients, particularly in elderly individuals, may contribute to this association.	[78,84,85]
	History of Abdominal Surgery	Strong adhesions from previous abdominal surgery may damage lymphatic vessels, increasing the risk of lymphocele.	[80,84]
	Deceased Donor Transplant	Inadequate lymphatic handling during donor nephrectomy may contribute to lymphocele formation.	[78]
	Rejection	Inflammation around the transplanted kidney may damage lymphatic vessels, promoting lymphocele formation.	[75]
Factors relevant to renal transplant surgery			
	Multiple Renal Arteries	The presence of multiple renal arteries requires more extensive lymphatic handling during anastomosis, increasing the risk.	[86]
	Inadequate Lymphatic Handling	Inadequate ligation of lymphatic vessels or damage due to electrocautery.	[87,88]
	Use of External Iliac Artery	Dense distribution of lymphatic vessels around the external iliac artery increases the risk.	[87,88]
	Placement of the Transplanted Kidney in the Left Iliac Fossa	Possible anatomical factors associated with the left iliac fossa may contribute to the risk.	[83]
Factors relevant to post-renal transplant management			
	Drain Placement	While drain placement may help prevent lymphocele formation, it may also increase the risk of lymphatic leakage.	[75,76,86,87,91]
	Immunosuppressive Therapy	High-dose steroid therapy, Tac, MMF, and mTOR inhibitors are associated with lymphatic complications.	[33,74,81,82,89,90]

### 3.2. Treatments

Mehrabi et al. recommend a grade-based treatment strategy for lymphocele and lymphorrhea management [92].

Grade A: Lymphoceles that do not require treatment or can be managed with diagnostic/therapeutic aspiration alone. These cases have minimal clinical impact and can be addressed non-invasively. However, aspiration carries a risk of recurrence.

Grade B: Lymphoceles requiring non-surgical intervention. If lymphatic leakage does not resolve spontaneously, additional treatments such as pharmacologic therapy (e.g., octreotide) or sclerotherapy may be necessary [93]. Cases with infection or complications requiring antibiotic therapy are also classified as Grade B. Sclerotherapy is a procedure in which a sclerosing agent is injected into the lymphocele after aspiration to induce adhesion of the cyst wall. Common sclerosing agents include povidone-iodine, ethanol, and tetracycline [88,94,95]. If infection is present or other treatment methods fail, drain placement may be necessary to facilitate lymphatic fluid drainage [96].

Grade C: Lymphoceles requiring surgical intervention, either by open or laparoscopic surgery. Procedures such as laparoscopic or open peritoneal fenestration are commonly performed, with reports indicating a low recurrence rate [79,97]. Lymphatic embolization is another treatment option in which embolic materials are injected into the lymphatic vessels under radiographic guidance to stop lymphatic leakage [95].

In recent years, minimally invasive treatments, such as aspiration and sclerotherapy, have become preferred over surgical interventions due to their lower risk of complications and shorter hospital stays. As a result, these methods are often considered the first-line treatment for lymphoceles [79,88]. Each treatment approach has its advantages and disadvantages, and selecting the optimal strategy requires careful discussion between the physician and the patient.

### 3.3. Preventive Strategies

Various preventive strategies have been explored to reduce lymphatic complications after kidney transplantation. However, each method has its own advantages and disadvantages, and the optimal approach has not yet been established.

Peritoneal fenestration is a technique that prevents lymphatic fluid accumulation by creating a connection between the peritoneal and retroperitoneal cavities [84]. The modified peritoneal fenestration technique, which combines fenestration with clip closure, has been evaluated for its safety and effectiveness in suppressing lymphocele formation, showing results comparable to conventional peritoneal fenestration [83]. Studies have demonstrated that peritoneal fenestration significantly reduces the incidence of lymphoceles. Ligation of lymphatic vessels using sutures has long been employed to prevent lymphatic leakage [75]. However, while studies comparing the bipolar vessel sealing system with silk ligation have reported no significant difference in the incidence of lymphatic complications, some reports suggest that sealing lymphatic vessels using a vessel-sealing system may result in the lowest incidence of post-transplant lymphoceles [87]. Minimally invasive transplant techniques have also been shown to limit lymphatic vessel injury, potentially reducing the occurrence of lymphoceles.

Povidone-iodine sclerotherapy involves injecting povidone-iodine through a drain to induce sclerosis of the lymphatic vessels [76]. While this method may reduce the incidence of lymphoceles, it carries the risk of tissue damage due to the sclerosing agent. Additionally, fibrin sealant has been reported to be effective in radical prostatectomy cases, a surgical field close to kidney transplantation [98].

Indocyanine green (ICG) fluorescence imaging has been introduced as a method for assessing lymph node and lymphatic vessel damage during surgery. Real-time intraoperative fluorescent lymphography using ICG has been reported, but due to the limited number of cases, further evaluation is required [99].

## 4. Infections (Urinary Tract Infections and Surgical Site Infection)

### 4.1. Urinary Tract Infections (UTIs)

#### 4.1.1. Etiology

The definition of UTI after kidney transplantation was established in 2019 by the American Society of Transplantation Infectious Diseases Community of Practice, classifying UTIs into four categories: asymptomatic bacteriuria (ASB), acute simple cystitis, acute pyelonephritis/complicated UTI, and recurrent UTI [100]. A recurrent UTI is defined as ≥3 UTI episodes within one year or ≥2 episodes within six months.

According to a report by Ariza-Heredia et al., the incidence rates were as follows: ASB (44%), acute simple cystitis (32%), acute pyelonephritis/complicated UTI (23%), and recurrent UTI (13%). Reports on the incidence of UTI after kidney transplantation vary widely due to differences in data collection methods, UTI definitions, data aggregation techniques, and clinical backgrounds. A review by Hollyer et al. reported that the incidence of post-transplant UTI was 33.5% (range: 7–80%), and the recurrence rate was 36% (range: 4–72%). Another meta-analysis study reported a prevalence of 35% for post-transplant UTIs [101,102]. When compared across different follow-up periods, the incidence of UTI was 20.6% within the first month post-transplantation and 30.2% between two and twelve months post-transplantation [103]. Other reports have demonstrated consistently high UTI prevalence regardless of follow-up duration, with rates of 34% and 43% at 1–2 years and 2–5 years post-transplantation, respectively [104].

In the general population, UTIs are known to be more common in females. However, reports on post-transplant UTIs show considerable variability in the proportion of female patients. A review reported that the proportion of female kidney transplant recipients with UTIs was 39.7%, suggesting a slightly lower prevalence in women [102]. In contrast, a meta-analysis by Wu et al. found that females had a significantly higher incidence of UTIs. The same study also examined potential regional differences but reported no significant difference in UTI prevalence between the United States and Europe [104].

Most UTIs occurring within the first six months after kidney transplantation are associated with ureteral stents, typically developing either during stent placement or within two weeks after stent removal [105]. After stent removal, the incidence of UTIs significantly decreases and becomes sporadic.

Acute graft pyelonephritis occurring within the first three months post-transplantation may negatively affect graft prognosis [106]. Some studies have reported that while serum creatinine levels temporarily increase during typical UTI episodes, they tend to recover over time [107]. However, patients with recurrent UTIs have been shown to have significantly lower estimated Glomerular Filtration Rate three years post-transplantation [108].

Recurrent UTIs may lead to renal cortical scarring, which could adversely impact graft function. While some studies have reported an association between UTIs and long-term renal function decline or increased mortality, others have found no clear correlation. Further research is needed to clarify this relationship [100,102,109].

In kidney transplant recipients, classical UTI symptoms such as frequent urination, dysuria, and urgency are often absent due to immunosuppression, surgical denervation of the transplanted kidney, or diabetes [110]. Consequently, UTIs may present as fever, urinary tract sepsis, or asymptomatic elevation of serum creatinine levels.

For recurrent UTIs, an evaluation of voiding function is necessary, particularly in diabetic patients, to assess for neurogenic bladder due to diabetic neuropathy and the presence of VUR [100].

In patients with long-term anuria due to dialysis, lower urinary tract symptoms (LUTS) are often under-evaluated before transplantation because of the absence of voiding activity. Once urination resumes after transplantation, previously unrecognized LUTS may become clinically apparent. In some cases, the development of UTI serves as an opportunity to detect these silent LUTS. Therefore, in anuric recipients, early postoperative evaluation of voiding function by urologists is recommended to identify and appropriately manage any underlying lower urinary tract dysfunction.

#### 4.1.2. Pathogens

The primary causative pathogens of UTIs after kidney transplantation are Gram-negative bacteria, similar to those observed in the general population without transplantation. The most commonly implicated pathogens include *Escherichia coli*, *Klebsiella* spp., *Pseudomonas aeruginosa*, and *Proteus mirabilis* [111].

Gram-positive bacteria can also cause UTIs, with *Enterococcus* spp. and *Staphylococcus aureus* being notable pathogens. In rare cases, fungi, viruses, parasites, and Mycoplasma may also be responsible for UTIs. Additionally, *Lactobacillus* spp., *Staphylococcus epidermidis* (a coagulase-negative staphylococcus), *Corynebacterium* spp., *Candida* spp., *Gardnerella vaginalis*, and non-β-hemolytic *Streptococcus* spp. are normally commensal organisms of the skin and vagina. These microorganisms can occasionally be detected in urine cultures due to shifts in microbial flora. While their presence does not necessarily indicate pathogenicity and is often of limited clinical significance, careful assessment is required in immunosuppressed patients to determine whether treatment is necessary [112,113,114,115]. In recent years, the prevalence of drug-resistant bacteria as causative agents of UTI in kidney transplant recipients has been increasing [102,116]. The rise in extended-spectrum β-lactamase-producing bacteria, multidrug-resistant *Pseudomonas aeruginosa*, carbapenem-resistant Enterobacteriaceae, vancomycin-resistant enterococci, and methicillin-resistant Staphylococcus aureus has become a significant public health concern [117]. This increasing prevalence complicates empirical treatment selection and restricts therapeutic options when drug-resistant pathogens are detected, making treatment more challenging.

#### 4.1.3. Risk Factors

A meta-analysis study involving 72,600 kidney transplant patients reported the following estimated risk factors for UTI: female sex (OR = 3.13; 95% CI: 2.35–4.17), older age (OR = 1.03; 95% CI: 1.00–1.05), history of UTI (OR = 1.31; 95% CI: 1.05–1.63), receiving a kidney from a deceased donor (OR = 1.59; 95% CI: 1.23–2.35), long-term use of an indwelling catheter (OR = 3.03; 95% CI: 1.59–6.59), ureteral stent placement (OR = 1.54; 95% CI: 1.16–2.06), diabetes (OR = 1.17; 95% CI: 0.97–1.41), hypertension (OR = 1.6; 95% CI: 1.26–2.28), acute rejection episodes (OR = 2.22; 95% CI: 1.45–3.4), and abnormal urinary tract anatomy (OR = 2.87; 95% CI: 1.44–5.74) [101]. A Cochrane review examining the relationship between immunosuppressants and UTI incidence found no significant difference in UTI occurrence MMF and azathioprine (AZA) users [118].

#### 4.1.4. Treatments

Early intervention is crucial in preventing the impact of UTIs on graft function [109]. Antimicrobial therapy should be selected based on local antimicrobial resistance patterns and clinical guidelines. Once urine and blood culture results are available, the initial antibiotic should be adjusted to a definitive-spectrum antibiotic to complete the treatment. In general patients, appropriate treatment and early transition from intravenous to oral antibiotics can shorten the duration of hospitalization [119]. However, due to the effects of immunosuppression, kidney transplant recipients may require a longer treatment duration compared to the general population [120]. If clinical improvement is insufficient, screening for complicated pyelonephritis should be performed. In cases of urinary tract obstruction, the placement of a catheter or stent is necessary to relieve the obstruction. If a renal abscess is present, drainage should be considered.

#### 4.1.5. UTI Prevention

According to the Cochrane Database of Systematic Reviews, the effectiveness of prophylactic perioperative antibiotics in preventing UTI after solid organ transplantation is very limited, with a relative effect of 0.88 (95% CI: 0.68–1.14) [121]. Furthermore, extending the duration of antibiotic administration did not demonstrate any significant benefit, with a relative effect of 0.49 (95% CI: 0.13–1.85). The current evidence supporting the use of prophylactic perioperative antibiotics in transplantation is of very low quality. Other measures to reduce the risk of UTI after kidney transplantation include the early removal of urinary catheters and ureteral stents, which are essential components of urinary tract management [122,123]. Additionally, pretransplant diabetes management may help lower the risk of UTI [124].

#### 4.1.6. Asymptomatic Bacteriuria (ASB) After Renal Transplantation

A prospective study of 209 renal transplant recipients followed for one year post-transplant, with urine cultures performed every three days for the first two weeks, weekly until one month, and at each outpatient follow-up visit, reported that 53% of subjects had at least one positive urine culture. Among these, 53% of bacteriuric episodes were asymptomatic, and 40% of patients had at least one episode of ASB [100].

A study by El-Amari et al. reported that even in untreated ASB cases, 57% resolved spontaneously, while only 59% resolved with antibiotic treatment [125]. Furthermore, 35% of treated cases developed antibiotic-resistant bacteria, suggesting that overuse of antibiotics contributes to the emergence of resistant strains. Other studies have also warned that the treatment of ASB increases the prevalence of resistant bacteria [126].

In a 36-month follow-up study of 189 renal transplant recipients, UTI developed in 51% of patients despite ASB treatment [127]. The highest incidence was immediately post-transplant (23%), followed by 10–17% within the first year and 2–9% after one year. Similarly, a retrospective study by Santithanmakorn et al. demonstrated that UTI negatively impacts graft function and survival in kidney transplant recipients [3]. However, ASB treatment within the first three months post-transplant did not reduce the incidence of UTI in the first year. Additionally, a recent systematic review and meta-analysis of randomized controlled trials confirmed that antibiotic treatment for ASB does not reduce the incidence of symptomatic UTI or improve graft function and overall patient outcomes [128]. Based on these findings, the Infectious Diseases Society of America Clinical Practice Guidelines for the Management of Asymptomatic Bacteriuria states the following: “In renal transplant recipients who had the renal transplant surgery >1 month prior, we recommend against screening for or treating ASB (strong recommendation, high-quality evidence). There is insufficient evidence to inform a recommendation for or against screening or treatment of ASB within the first month following renal transplantation.” [114].

### 4.2. Surgical Site Infections (SSIs)

#### 4.2.1. Etiology

The incidence of SSI following kidney transplantation has been reported to have a median rate of 3.7% (range: 1.7–18.5%) [129,130,131,132,133,134,135,136,137,138]. Compared to other solid organ transplants, the incidence tends to be lower [117]. However, in pancreas–kidney transplantation, the incidence of SSI increases to 24.3% [139].

The diagnosis of SSI is based on clinical symptoms, including purulent discharge, erythema, swelling, and pain, in conjunction with microbiological culture results. The majority of SSIs occur within the first 30 days post-transplant, with an incidence rate of 4.19 per 100 person–months [4,133,135,136,140]. Patients who develop SSIs experience a prolonged hospital stay by an average of 26.5 days, an increase in hospital costs by $24,454, and a reduction in hospital profit by $4278 [133,135]. Furthermore, SSI significantly decreases both patient survival and graft survival in transplant recipients [4,141,142].

#### 4.2.2. Causative Pathogens

The most commonly isolated pathogens are Staphylococcus aureus, coagulase-negative staphylococci, and enterococci [4,117,133]. Some studies have reported coagulase-negative staphylococci as the most frequently isolated species; however, this may be influenced by contamination during specimen collection. Gram-negative bacteria are also implicated, including *Escherichia coli*, *Klebsiella species*, *Enterobacter species*, *Pseudomonas aeruginosa*, and *Acinetobacter baumannii* [131,132,133]. In some cases, multiple bacterial species are detected, and yeast, particularly the *Candida species*, may also be isolated [143,144].

#### 4.2.3. Risk Factors

Multiple factors contribute to the development of SSI, which can be categorized into recipient, donor, and surgical-related factors (Table 2).

Recipient-Related Factors: Patients with a BMI exceeding 27 kg/m^2^ have an increased risk of SSI [130,135,142,145]. Additionally, for each 1 kg/m^2^ increase in BMI, the risk of SSI is estimated to increase by 6% [146]. Diabetes and advanced age are also associated with a higher risk of SSI [130,135,139,142]. Furthermore, recipients with a history of narcotic use or smoking have been reported to be at an increased risk of SSI [146,147]. Malnutrition may also increase the risk of surgical site infection (SSI) [142]. The occurrence of DGF is associated with a higher risk of SSI [117,130,135,141]. The use of immunosuppressive drugs increases the overall risk of infections, with mTOR inhibitors specifically linked to delayed wound healing and a potentially higher risk of SSI [89,134,148]. Excessive immunosuppression further predisposes patients to SSI [134]. The use of MMF, AZA [117,141], and antithymocyte globulin or basiliximab [149] has also been reported to increase SSI risk. Additionally, preoperative antibiotic use may elevate the risk of infections caused by multidrug-resistant organisms [141].

Donor-Related Factors: If the donor carries colonizing bacteria or has an active infection, the risk of SSI increases, particularly in the case of Gram-negative bacteria [117]. Microbial contamination of the organ preservation solution is associated with an increased risk of SSI [117,141,150]. Fungal contamination, particularly by the *Candida species*, is of particular concern [144]. Kidney transplantation from a deceased donor has been reported to carry a higher risk of SSI compared to transplantation from a living donor [130,135].

Surgical-Related Factors: Surgical risk factors for SSI include prolonged operative time, blood transfusion, hematoma formation, reoperation, urine leakage, and lymphorrhea [89,141,142,146,148].

#### 4.2.4. Treatments

Early diagnosis and appropriate antibiotic therapy are fundamental in SSI management [117,133,140]. Antibiotics should be selected based on the identified causative pathogen and its antimicrobial susceptibility determined by culture testing. Surgical drainage is necessary if an abscess has formed. Debridement (wound cleansing) is performed to remove infected or necrotic tissue. mTOR inhibitors (e.g., sirolimus, everolimus) may delay wound healing; therefore, temporary discontinuation of these drugs may be considered during treatment [117,141,148]. Negative pressure wound therapy is often used in conjunction with these treatments to remove exudates and infectious materials from the wound, promote the formation of granulation tissue, and accelerate wound healing [140].

#### 4.2.5. Prevention of SSI

According to the Cochrane Database of Systematic Reviews, the effectiveness of prophylactic antibiotics in preventing SSI during the perioperative period of solid organ transplantation is reported to be relatively poor, with a relative effect of 0.42 (0.21–0.85) [121]. Moreover, this effect remained unchanged regardless of the duration of antibiotic administration. A RCT comparing vancomycin plus ceftriaxone with cefazolin monotherapy for perioperative prophylaxis found no significant difference in postoperative SSI incidence. These findings suggest that the use of multiple antibiotics or broad-spectrum antibiotics does not contribute to reducing SSI risk [151]. Based on these results, clinical guidelines recommend the use of a first-generation cephalosporin for perioperative antibiotic prophylaxis, administered for 24 h or less in renal transplantation [14,117,141].

## 5. Vascular Complications

### 5.1. Etiology

The incidence of vascular complications in kidney transplant recipients is reported to be approximately 10%. These complications include renal artery stenosis, arterial thrombosis, venous thrombosis, arterial anastomotic aneurysms, and arterial dissection [31,33,152,153]. Among these, renal artery stenosis and thrombosis account for 50–80% of vascular complications [154,155,156]. Vascular complications typically occur in the early postoperative period [155]. Renal artery stenosis reduces renal allograft perfusion, leading to ischemic injury and potentially causing long-term renal dysfunction. The frequency of stenosis formation varies depending on whether the renal artery is anastomosed to the internal iliac artery or external iliac artery, with a higher frequency reported in external iliac artery anastomosis [152]. The use of a Carrel patch in vascular anastomosis has been discussed in relation to transplant renal artery stenosis (TRAS) [157,158]. It is not the patch itself but rather the presence of atherosclerotic plaques within the Carrel patch—derived from the donor artery—that has been associated with an increased risk of TRAS. These plaques may contribute to luminal narrowing, endothelial injury, thrombosis, and fibrosis, all of which can lead to stenosis. Therefore, when using grafts from donors with a high risk of atherosclerosis, evaluation of the Carrel patch for plaque is essential, and excision may be considered if significant lesions are found. TRAS commonly occurs within three months post-transplant, and vascular interventions are typically performed around four months after transplantation [29,159]. Anastomotic strictures (58.4%) are more common than non-anastomotic strictures (41.5%). Arterial and venous thrombosis can completely obstruct blood flow to the transplanted kidney, potentially leading to acute kidney failure and, in the worst cases, graft loss.

#### 5.1.1. Risk Factors for Vascular Complications

Anatomical abnormalities of the donor renal vasculature, particularly the presence of multiple arteries, may increase the risk of vascular complications [160,161]. On the recipient side, underlying conditions such as diabetes, hypertension, and hyperlipidemia may contribute to a higher risk of vascular complications [156,157,159]. Certain antibody induction therapies have been reported to induce a transient hypercoagulable state due to cytokine release, which may increase the risk of renal artery thrombosis in the transplanted kidney [162]. A history of previous kidney transplantation, hypertension, and diabetes in the recipient also increases the risk [154,157,159]. Severe aortoiliac atherosclerosis can affect the anastomotic site of the transplant vessel, impairing blood flow and increasing the risk of TRAS and thrombosis [163,164]. Additionally, vascular endothelial inflammation and damage caused by rejection may further promote the development of TRAS [164].

#### 5.1.2. Treatment

Transplant renal artery stenosis (TRAS) occurring within the first 10 days after transplantation is relatively rare, but it may result from surgical factors, preexisting recipient vascular disease, or arterial dissection. In such cases, endovascular treatment is generally the first-line approach. However, when technical issues during surgery are suspected to be the cause, surgical revision should also be considered [165,166]. For late-onset TRAS, percutaneous transluminal renal angioplasty (PTRA) is performed using a balloon catheter to dilate the stenotic vessel, and in some cases, percutaneous transluminal renal angioplasty with stenting (PTRAS) is used to place a stent at the same site. These interventional therapies have been shown to significantly reduce blood pressure and serum creatinine levels compared to preoperative conditions [167]. Furthermore, no significant differences in the incidence of graft failure or restenosis have been reported between these two methods [159]. While many reports indicate improved graft function and favorable long-term outcomes, it has also been observed that patients with TRAS experienced a faster decline in renal function over an average follow-up period of 43.6 months compared to those without TRAS [157]. Potential complications of PTRA and PTRAS include arterial dissection, arterial perforation, hematoma formation, and restenosis; however, these occur at a low incidence, and the procedures are considered safe, providing an alternative to open surgery [33,163].

For arterial and venous thrombosis, thrombolytic therapy and percutaneous thrombectomy are performed. If these methods are ineffective, surgical thrombectomy may be required [152,155]. In cases of arterial aneurysms, endovascular therapy or surgical repair is performed.

#### 5.1.3. Prevention

During donor kidney procurement and transport, meticulous care must be taken to avoid direct mechanical injury to the renal artery. In deceased donor transplantation, the use of an aortic cuff for arterial anastomosis has been reported to reduce the incidence of anastomotic stenosis [168]. Additionally, longer warm ischemia time, cold ischemia time, and total ischemia time have been associated with the development of TRAS [161]. Ischemia–reperfusion injury may cause oxidative stress and endothelial damage, underscoring the importance of minimizing the time from graft retrieval to reperfusion in the recipient [157]. During surgery, meticulous vascular anastomosis is essential to minimize the risk of vascular endothelial injury and twisting. Careful surgical techniques can help prevent vascular complications [33]. Additionally, intraoperative and postoperative Doppler ultrasound is a valuable tool for assessing blood flow and detecting early signs of vascular compromise [31].

## 6. Neurologic Complications

One of the most common perioperative neurological complications is femoral nerve neuropathy, which results from damage or compression of the femoral nerve [169,170]. Clinical symptoms include sensory impairment in the anterior thigh and medial lower leg, difficulty in knee extension, quadriceps muscle weakness, and diminished or absent patellar reflex. A study by Li et al., which examined 1830 kidney transplant recipients, identified 83 cases of femoral nerve palsy, reporting that its incidence was associated with the selection of iliac arteries, the duration of arterial anastomosis, and injury to the iliolumbar or deep iliac circumflex artery [171]. Symptoms typically appear immediately after surgery. The primary cause of femoral nerve dysfunction following kidney transplantation is compression by self-retaining retractors used during surgery, particularly when the blades exert excessive pressure on the iliopsoas muscle and femoral nerve. Additional causes include compression from hematoma, ischemia, and suture-related nerve entrapment [172]. In most cases, femoral nerve injury resolves spontaneously, though recovery may take several weeks to months [173,174,175]. While motor dysfunction tends to improve more rapidly, sensory impairment may persist for an extended period [176].

Neurological complications following kidney transplantation can also arise from neurotoxicity induced by immunosuppressive agents. Tac, a commonly used immunosuppressant, has been reported to cause tremor, Posterior Reversible Encephalopathy Syndrome, and optic neuropathy, among other neurological toxicities [177,178,179]. Proper postoperative monitoring of tacrolimus blood levels is essential for prevention. If neurotoxicity occurs, dose reduction or switching to an alternative calcineurin inhibitor may be necessary.

Infectious neurological complications can also occur in kidney transplant recipients due to opportunistic infections such as cytomegalovirus and herpesviruses [180,181,182,183]. Additionally, fungal infections such as Cryptococcus can lead to meningitis or brain abscesses and should be considered in the differential diagnosis [33,172,184].

## 7. Incisional Hernia

### 7.1. Etiology

A meta-analysis of 16,018 patients reported that the pooled incidence of incisional hernia (IH) after kidney transplantation was 4% (CI 3–5%) [185]. Additionally, a systematic review found a time-dependent increase in IH incidence, with rates of 2.5% at 1 year, 4.9% at 5 years, and 10% at 10 years after kidney transplantation [186].

### 7.2. Risk Factors

The risk factors for incisional hernia can be classified into patient-related factors and transplant-related factors (Table 3). Patient-related factors include obesity, advanced age, smoking, pulmonary disease, and prolonged dialysis duration [96,185,187,188,189,190]. Transplant-related factors include the type of immunosuppressive therapy, the choice of incision, surgical technique, history of previous surgeries, and the presence of perioperative complications [96,185,191,192,193,194]. To reduce the risk of incisional hernia after kidney transplantation, several preventive measures can be taken. For patient-related factors, obese patients should be encouraged to lose weight and quit smoking before transplantation. Regarding transplant-related factors, selecting an appropriate incision is essential. Some studies suggest that the hockey-stick incision, commonly used in kidney transplantation, is associated with a higher incidence of incisional hernia compared to oblique incisions [185,194]. However, these findings are based on limited retrospective studies. In general abdominal surgery, the risk of incisional hernia is considered similar across different incision types, with incision size being a more significant determinant of hernia formation than the type of incision itself [195]. Thus, minimizing the incision size may be a key factor in reducing the risk of incisional hernia [196].

In addition, in general surgery, various techniques have been reported to reduce the risk of incisional hernia, including shortening the surgical time, opting for continuous fascial suturing, maintaining a suture interval of 5–8 mm, and ensuring proper control of SSIs [197]. Furthermore, clinical trials are investigating the prophylactic placement of mesh during wound closure, which may help reduce the incidence of incisional hernia following kidney transplantation [198].

### 7.3. Treatment

The primary treatment for incisional hernia following kidney transplantation is surgical repair. A surgical approach was chosen in 61% (CI: 14–100%) of cases, with a reported recurrence rate of 16% (CI: 9–23%) after repair [185]. Laparoscopic surgery offers several advantages over open surgery, including smaller incisions and faster postoperative recovery [199,200].

Surgical repair commonly involves the use of mesh, which is recommended to provide safe and effective treatment while minimizing the risk of recurrence [96]. The choice of mesh type plays a crucial role in the repair process. Polypropylene mesh, known for its high tensile strength, is widely used but carries an increased risk of infection [201]. In contrast, the absorbable mesh is gradually resorbed by the body, presenting a lower infection risk but offering less durability compared to polypropylene mesh.

### 7.4. Conclusions

In conclusion, surgical and infectious complications remain a major challenge in kidney transplantation, with significant implications for graft survival and patient outcomes. Early detection, risk stratification, and timely intervention are critical for reducing morbidity and improving long-term prognosis. By understanding the multifactorial nature of these complications and integrating evidence-based practices, transplant teams can optimize outcomes and improve the quality of life for kidney transplant recipients. Further research is warranted to refine prevention and treatment strategies for specific complications in diverse clinical settings.

## Figures and Tables

**Table 2 jcm-14-03307-t002:** Risk factors for SSI.

	Risk Factors	Description	Reference
Recipient-Related Factors			
	Diabetes mellitus	Diabetes-induced delayed wound healing increases the risk of SSI.	[130,135]
	Obesity	The risk of SSI increases in patients with a BMI >27 kg/m^2^.	[130,135,142,145,146]
	Advanced Age	Advanced age may contribute to an increased risk of SSI due to reduced healing capacity.	[139,142]
	History of past narcotic use		[146]
	History of smoking		[147]
	Malnutrition	Malnutrition may cause delayed wound healing, thereby increasing the risk of SSI.	[142]
	DGF	The occurrence of DGF is associated with a higher risk of SSI.	[117,130,135,141]
	The use of immunosuppressive drugs	Particularly, the use of mTOR inhibitors delays wound healing and increases the risk of SSI.	[89,117,134,141,148,149]
	Preoperative use of antibiotics	Preoperative use of antibiotics may increase the risk of SSI caused by multidrug-resistant organisms.	[141]
Donor-Related Factors			
	Donor-Colonized or Infected Pathogens	Gram-negative bacilli transferred from the donor can be a significant contributing factor to the development of SSI.	[117]
	Contamination of Organ Preservation Solution	Contamination of the organ preservation solution with microorganisms, such as *Candida species*, increases the risk of SSI.	[117,141,147,150]
	Deceased Donor	Kidney transplantation from a deceased donor carries a higher risk of SSI compared to transplantation from a living donor.	[130,135]
Surgical-Related Factors			
	prolonged operative time, reoperation, blood transfusion, hematoma formation, urine leakage, and lymphorrhea		[89,141,142,146,148]

**Table 3 jcm-14-03307-t003:** Risk factors for incisional hernia.

	Risk Factors	Description	Reference
Patient-Related Factors			
	Obesity	Obesity is a significant risk factor for incisional hernia, as excess abdominal fat increases stress on the abdominal wall.	[185,187,188,189]
	Advanced Age	As patients age, their tissue healing capacity declines, leading to an increased risk of incisional hernia.	[187,190]
	Smoking	Smoking causes vasoconstriction and reduces blood flow to tissues, which delays wound healing and increases the risk of incisional hernia.	[96,187,189]
	Pulmonary Disease	Patients with pulmonary disease may experience frequent coughing, which increases intra-abdominal pressure and the risk of incisional hernia.	[185]
	Dialysis Duration	Prolonged dialysis duration is an independent risk factor for incisional hernia.	[190]
Transplant-Related Factors			
	Immunosuppressive Therapy	Immunosuppressive medications, which are essential for preventing rejection, may delay wound healing and increase the risk of incisional hernia. In particular, MMF and mTOR inhibitors are associated with an increased risk of hernia formation.	[96,185,188,191,192,193]
	Type of Incision	For kidney transplantation, oblique and hockey-stick incisions are commonly used. Some studies suggest that hockey-stick incisions are associated with a higher incidence of incisional hernia.	[185,194]
	Surgical Technique	Robotic-assisted kidney transplantation has been reported to have a lower incidence of incisional hernia compared to traditional open surgery.	[185]
	Previous Surgeries	The risk of incisional hernia is higher in areas with previous surgeries, as scar tissue formation weakens tissue integrity.	[187]
	Complications	Surgical site infection, delayed graft function, and lymphocele can increase the risk of incisional hernia by impairing wound healing or weakening the abdominal wall.	[96,185,188,193]

## Data Availability

This review article did not generate any new data.

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
