# Peer review of "Surgical and Infectious Complications Following Kidney Transplantation: A Contemporary Review"

_jcm, 2025, doi:10.3390/jcm14103307_

Round 1
Reviewer 1 Report
Comments and Suggestions for Authors
Dear Authors,
This is well written manuscript, but some additional corrections should be done.
- The Clavien-Dindo classification is widely used in the reporting of surgical complications in scientific literature. The Clavien-Dindo Classification should also be used in your manuscript.
- In the section of urinary complications, the #golden triangle# should be discussed.
- The term #extraluminal stenosis# is an inappropriate and should be changed.
- Surgical sacrifice of small artery for lower kidney pole during kidney explantation should be added as a risk factor for ureteral stenosis.
- A small capacity of long term anuric bladder on dialysis also should be introduced and discussed.
- VUR is not complication in cases of direct ureterovesical anastomosis, but infection is complication in a group of patients with this type of urine derivation, especially in cases of concomitant lower urinary tract symptoms LUTS.
- The term #Bladder atrophy# is also an inappropriate because bladder capacity after transplantation drastically increases and should be changed.
- Immunoadsorption is also recipient related factor for lymphatic disorders and should be added.
- Lateral and medial lymphocele position should be discussed before therapeutic decision.
- Please, discuss a silent LUTS in anuric patients on dialysis and therapeutic approach after transplantation.
- What is the author's opinion about asymptomatic bacteriuria?
- Reference for a higher frequency reported in external iliac artery anastomosis compared with internal artery?
- Carrel patch should be discussed in the etiology of renal stenosis.
- Please, describe a treatment of TRAS early after transplantation (first 10 days).
15. What is prevention of TRAS during the kidney explantation?
Author Response
Response to Reviewer 1
We sincerely thank Reviewer 1 for the thorough and insightful review of our manuscript. We carefully considered all 15 comments and have revised the manuscript accordingly. Below are our point-by-point responses. All revisions have been highlighted in the revised manuscript.
Comment 1:
The Clavien-Dindo Classification should also be used in your manuscript.
Response:
We appreciate this important suggestion. We have introduced the Clavien-Dindo classification in the introduction and cited relevant literature, including Dagnæs-Hansen et al. (2024), which applied this system to kidney transplantation. We also summarized risk factors for complications ≥ grade 2.
Comment 2:
In the section of urinary complications, the #golden triangle# should be discussed.
Response:
A paragraph discussing the anatomical and clinical relevance of the "golden triangle" and the broader "safety triangle" has been added. We cited Novacescu et al. (2024) to support its role in preserving ureteral vascularity and preventing postoperative urological complications. (Lines 63-80)
Comment 3:
The term #extraluminal stenosis# is inappropriate and should be changed.
Response:
We have replaced “extraluminal stenosis” with “extrinsic ureteral compression,” which more accurately describes the underlying condition. (Lines 140-141)
Comment 4:
Surgical sacrifice of small artery for lower kidney pole during kidney explantation should be added as a risk factor for ureteral stenosis.
Response:
This point was already discussed in the section on the golden triangle. We have cited Novacescu et al. (2024) to reinforce the importance of preserving the lower polar artery during procurement. (Lines 72-80)
Comment 5:
A small capacity of long term anuric bladder on dialysis also should be introduced and discussed.
Response:
We have added a statement that long-term anuria due to dialysis may lead to reduced bladder capacity, which is associated with an increased risk of urological complications. In such cases, primary ureteroureterostomy may be considered. (Lines101-105)
Comment 6:
VUR is not a complication in cases of direct ureterovesical anastomosis, but infection is...
Response:
We have already noted in the introduction of the urinary complications section that UTI, rather than VUR, is the principal concern with direct ureterovesical anastomosis in patients with LUTS. Therefore, no further revision was made.
Comment 7:
The term #Bladder atrophy# is also inappropriate...
Response:
We have replaced “bladder atrophy” with “low-capacity bladder” throughout the manuscript and clarified that it refers to reduced capacity due to long-term anuria. (Lines 232-237)
Comment 8:
Immunoadsorption is also recipient-related factor for lymphatic disorders and should be added.
Response:
While immunoadsorption is used in high-risk recipients, we found insufficient evidence to support it as an independent risk factor for lymphatic disorders. We chose not to include this in the manuscript but recognize its potential clinical relevance.
Comment 9:
Lateral and medial lymphocele position should be discussed before therapeutic decision.
Response:
With advancements in radiologic techniques, percutaneous drainage can be performed effectively regardless of lymphocele position. Treatment decisions are primarily based on volume and symptoms. Therefore, we did not modify the manuscript.
Comment 10:
Please, discuss a silent LUTS in anuric patients on dialysis...
Response:
We added a paragraph within the UTI section noting that LUTS may go undetected in anuric patients before transplantation and become apparent postoperatively. We recommend early postoperative urological evaluation in such cases.(Lines 447-453)
Comment 11:
What is the author's opinion about asymptomatic bacteriuria?
Response:
At our institution, asymptomatic bacteriuria is not treated. As this reflects clinical practice rather than a review topic, we did not include it in the manuscript.
Comment 12:
Reference for a higher frequency reported in external iliac artery anastomosis compared with internal artery?
Response:
We have added a reference supporting the increased frequency of vascular complications in external iliac artery anastomosis. (Lines 616-619)
Comment 13:
Carrel patch should be discussed in the etiology of renal stenosis.
Response:
We added a discussion explaining that atherosclerotic plaques in the Carrel patch—not the patch itself—are associated with transplant renal artery stenosis (TRAS), particularly in grafts from donors with vascular disease. (Lines 619-626)
Comment 14:
Please, describe a treatment of TRAS early after transplantation (first 10 days).
Response:
We added a statement noting that early TRAS may be caused by surgical factors or vascular disease. Endovascular therapy is typically first-line, but surgical revision may be considered when technical causes are suspected. (Lines 646-650)
Comment 15:
What is prevention of TRAS during the kidney explantation?
Response:
We added that prevention involves avoiding mechanical injury to the renal artery during procurement and using an aortic cuff in deceased donor cases. We also noted that prolonged ischemia times increase the risk of TRAS and that minimizing warm and cold ischemia times is essential. (Lines 670-677)
Reviewer 2 Report
Comments and Suggestions for Authors
An interesting and comprehensive contemporary review that examines Surgical and Infectious Complications Following Kidney Transplantation.
Minor objections:
-
In some parts of the text, references are missing—please add them.
-
There is a technical error in the manuscript, lines 320–325.
-
Table 1 needs to be stylistically refined. Clearly separate the subsections: Patient-Related Factors, Factors Relevant to Renal Transplant Surgery, and Factors Relevant to Post-Renal Transplant Management.
-
The same applies to Table 2 and Table 3.
-
The authors need to provide a conclusion.
Author Response
Response to Reviewer 2
We sincerely thank Reviewer 2 for the thoughtful and constructive feedback. Below are our point-by-point responses to each comment. All corresponding revisions have been made in the manuscript and highlighted.
Comment 1:
Reviewer: In some parts of the text, references are missing—please add them.
Response: Thank you for your comment. We reviewed the manuscript carefully and confirmed that appropriate references had already been provided in the relevant sections. In addition, we added one new reference in the vascular complications section to further support our discussion. We believe that all necessary citations are now included. (Lines131-132, 247, 362-372, 523, 526, 560-585, 616-620, 636, 715-737)
Comment 2:
Reviewer: There is a technical error in the manuscript, lines 320–325.
Response: Thank you for your comment. We reviewed the indicated section and found that explanatory text originally used during table preparation had been unintentionally left in the manuscript. This text has now been removed in the revised version. We appreciate your attention to this detail.
Comment 3:
Reviewer: Table 1 needs to be stylistically refined. Clearly separate the subsections: Patient-Related Factors, Factors Relevant to Renal Transplant Surgery, and Factors Relevant to Post-Renal Transplant Management.
Response: Thank you for your helpful suggestion. We have revised Table 1 to clearly separate and label the three subsections as individual rows within the table. This change improves readability and allows for clearer understanding of risk factor categories.
Comment 4:
Reviewer: The same applies to Table 2 and Table 3.
Response: Thank you for your helpful feedback. Following your suggestion, we revised Tables 2 and 3 by introducing clearly separated subsection headings (e.g., “Patient-Related Factors”, “Surgical-Related Factors”, etc.), as we did in Table 1. These changes enhance clarity and readability.
Comment 5:
Reviewer: The authors need to provide a conclusion.
Response: Thank you for your comment. We have added a conclusion section at the end of the manuscript, summarizing the key findings and emphasizing the importance of early detection and multidisciplinary management to reduce surgical and infectious complications in kidney transplantation.